# The Case for Evaluating Causal Models Using Interventional Measures and Empirical Data

**Amanda Gentzel, Dan Garant, and David Jensen**
College of Information and Computer Sciences
University of Massachusetts Amherst

## Abstract

Causal modeling is central to many areas of artificial intelligence, including complex reasoning, planning, knowledge-base construction, robotics, explanation, and fairness. An active community of researchers develops and enhances algorithms that learn causal models from data, and this work has produced a series of impressive technical advances. However, *evaluation techniques* for causal modeling algorithms have remained somewhat primitive, limiting what we can learn from experimental studies of algorithm performance, constraining the types of algorithms and model representations that researchers consider, and creating a gap between theory and practice. We argue for more frequent use of evaluation techniques that examine *interventional measures* rather than structural or observational measures, and that evaluate using *empirical data* rather than synthetic data. We survey the current practice in evaluation and show that the techniques we recommend are rarely used in practice. We show that such techniques are feasible and that data sets are available to conduct such evaluations. We also show that these techniques produce substantially different results than using structural measures and synthetic data.

## 1 Introduction

Evaluation is central to research in artificial intelligence and machine learning [Cohen, 1995, Langley, 2011]. How we evaluate algorithms determines our perception of the relative effectiveness and usefulness of different approaches, and this knowledge guides choices about future research directions. As Cohen and Howe [1989] explained three decades ago: "Ideally, evaluation should be a mechanism by which AI progresses both within and across individual research projects. It should be something we do as individuals to help our own research and, more importantly, on behalf of the field."

As fields develop, protocols for evaluation need to develop alongside them. In this paper, we offer an empirical analysis of the set of techniques typically used to evaluate algorithms for learning causal models, and we show that this set could be substantially enhanced. The ultimate goal of most algorithms for causal modeling is to learn models capable of accurately estimating the effects of interventions in real-world systems. With this goal in mind, we would like to evaluate algorithms by comparing their estimates to actual interventional effects on data produced by a real-world system. In practice, though, many evaluations fall short of this ideal, most frequently using only synthetic data and structural or observational measures. Without the use of *empirical data*, our evaluations produce little information about whether our algorithms generalize to real-world systems, and this greatly reduces their likelihood of widespread adoption by others outside of the field. Without the use of *interventional measures*, our evaluations produce little information about whether learned models will accurately estimate the effects of interventions, limiting their real-world utility.

Note that we do not argue for *replacing* the prevailing techniques for evaluation. These techniques have substantial value, both in assessing overall performance and in allowing fine-grained experiments

to diagnose specific performance issues. Rather, we argue for *augmenting* the current suite of evaluation techniques to gather experimental evidence that the prevailing techniques cannot. We also do not contend that interventional measures and empirical data are entirely absent from current studies. A very small minority of recent studies use these techniques in combination. Rather, we argue that interventional measures and empirical data should be used routinely, and should be used in combination, for any serious study of algorithms for learning causal models. Indeed, the conclusions of most studies that lack such evaluation techniques should be considered exploratory and would benefit from additional evaluation.

We make the following contributions:

**C1 Decomposition of Evaluation Techniques.** We decompose evaluation techniques into three interacting components: the data source, the algorithm, and the evaluation measure, allowing for a modular discussion of the interacting components of an evaluation.

**C2 Survey of Current Techniques.** We provide a detailed survey of recent literature in causal modeling to provide a quantitative understanding of current evaluation practices.

**C3 Critique of Current Practice.** We provide evidence that increased adoption of both empirical data and interventional measures would be beneficial to the community.

## 2 Survey of Current Techniques

To assess how frequently different evaluation techniques are used in practice, we surveyed recent computer science publications on causal modeling. We collected papers from the past five UAI, NeurIPS, AAAI, ICML, and KDD conferences, as well as causality workshops held at UAI. We examined papers whose titles contained the terms 'cause', 'causal', or 'causality' and then narrowed this selection of papers to those that describe, propose, or evaluate a causal modeling algorithm. This resulted in a final set of 111 papers, of which 82% (91) reported any sort of evaluation.[1] Citations to all 111 papers are provided in the Supplementary Material.

The counts of papers included in the final survey are shown in Table 1. While some relevant papers may fall outside of our search parameters, this approach captures a reasonably representative sample of recent work within computer science on causal modeling, allowing us to infer which techniques are used in practice and how frequently these techniques are used.

Table 1: Papers included in the survey

| Venue | 2014 | 2015 | 2016 | 2017 | 2018 | **Total** |
|---|---|---|---|---|---|---|
| UAI | 2 | 3 | 5 | 3 | 7 | 20 |
| NeurIPS | 3 | 5 | 4 | 6 | 13 | 31 |
| AAAI | 1 | 6 | 2 | 4 | 5 | 18 |
| ICML | 1 | 5 | 1 | 3 | 5 | 15 |
| KDD | 0 | 2 | 3 | 0 | 2 | 7 |
| UAI-W | 2 | 2 | 4 | 3 | 9 | 20 |
| **Total** | 9 | 23 | 19 | 19 | 41 | 111 |

Table 2: Number of papers using different evaluation techniques

| | | **Data Sources** | |
|---|---|---|---|
| | | Synthetic | Empirical |
| **Evaluation Measures** | Structural | 44 | 23 |
| | Observational | 22 | 14 |
| | Interventional | 11 | 6 |
| | Visual Inspection | 0 | 19 |

### 2.1 Survey Results

For ease of exposition, we decompose evaluation techniques into three components: (1) the data source; (2) the algorithm under evaluation; and (3) the evaluation measure. These dimensions are highly dependent—a choice of one can determine feasible choices for the others. For example, models learned from observational macro-economic data often cannot be compared against a known structure because there exists no ground truth, and models consisting only of non-parameterized structure cannot be compared to interventional effects because the models cannot produce such estimates.

**Data Sources.** The surveyed papers used a wide range of data sources, but they fall into two broad categories: synthetic and empirical. We categorized data as empirical when it was collected from

a "real world" system, whether that was a randomized clinical trial, a global financial system, or user interaction with a website. The important distinction is that empirical data was collected from a process or a system that exists for some purpose beyond scientific research. Synthetic data includes anything else, including data generated from a randomly instantiated directed graphical model or from a simulation intended to reflect a real-world system. In our survey, we found many examples of both, and while synthetic data is used more frequently, both are still common. 81% (74) of papers surveyed used synthetic data, 67% (61) used empirical data, and 48% (44) used both.

**Algorithms.** The algorithm under evaluation is not part of the evaluation technique *per se*, but aspects of the algorithm strongly influence how evaluation can, and should, be performed. Algorithms fall into two broad categories, bivariate and multivariate, based on the number of variables they consider, although there are many variants.

Some bivariate algorithms infer only the direction of effect (whether $A$ causes $B$ or $B$ causes $A$). Others estimate the magnitude of effect between treatment and outcome, while adjusting for the effects of a number of covariates. Bivariate methods include Granger causality analysis [Granger, 1969], additive noise models [Peters et al., 2014], and analyses that use the potential outcomes framework [Rubin, 2005]. The most common variety of multivariate algorithm learns a directed acyclic graph (DAG). Multivariate algorithms are significantly more prevalent in the data, accounting for 60% (55/111) of papers surveyed. Bivariate algorithms account for 30% (34/111) of papers surveyed, split between those focused on orientation (10%), magnitude of effect (15%), or both (5%). The remaining papers in the survey fall in between, including those that aim to determine the joint effect of multiple treatment variables on a single outcome.

**Evaluation Measures.** At the heart of any evaluation technique is a measure of performance. At a high level, evaluation measures fall into two categories: structural and distributional. Structural measures include all measures designed to assess whether the structure (including both existence of edges and edge orientation) learned by the algorithm matches the ground truth. Structural measures include structural Hamming distance (SHD), precision, recall, F1-score, true-positive rate, area under the ROC curve (AUROC), and structural intervention distance (SID) [Peters and Bühlmann, 2015].

Distributional measures capture how well the algorithm can estimate quantitative dependence. Such measures can be further subdivided into observational and interventional measures. Observational measures compare the learned distribution with an observational ground truth (i.e. probability queries which do not involve a *do* operator). This could be a measure of individual edge strengths in a directed graphical model or a measure of the error when predicting a given outcome variable. Interventional measures, on the other hand, compare the learned distribution to ground truth obtained through intervention. Common interventional measures include KL-divergence, total variation distance, and measures of average and conditional treatment effect.

Of the types of evaluation measures, structural measures are the most common, being used in 55% (50) of papers surveyed. Distributional measures are slightly less common, being used in 46% (42) of papers. The vast majority of the distributional measures used, however, are observational rather than interventional; observational measures are used in 32% (29) of papers, while interventional measures are used in only 14% (13).

The choice of evaluation measure depends on both the data generating process and type of algorithm, which is reflected in our survey. When synthetic data is evaluated, structural measures are used 59% (44/74) of the time. However, when empirical data is evaluated, structural measures are used only 38% (23/61) of the time, since empirical data is less likely to have ground truth. This lack of ground truth sometimes prevents any significant evaluation when using empirical data—26% (16/61) of empirical evaluations used only visual inspection of the results, with no ground truth. Table 2 summarizes the interaction between data source and evaluation measure in the survey.

## 2.2 Findings

The survey makes clear that the vast majority of papers that perform evaluation use either (1) synthetic data; or (2) empirical data combined with non-interventional measures (observational measures, structural measures, or visual inspection). Our proposed ideal evaluation (empirical data and interventional measures) is used in only 7% (6) of papers. This raises an obvious question: Are the most commonly used evaluation techniques sufficient for determining whether algorithms for learning causal models will work effectively in realistic scenarios? As we argue below, they are not.

# 3 The Case for Empirical Data

As already noted, nearly all causal modeling algorithms are ultimately designed for use outside of a laboratory, on real systems to infer useful causal knowledge about the world. Despite this, evaluation of such algorithms often uses synthetic rather than empirical data.

## 3.1 Limitations of Synthetic Data

Researchers have developed several approaches to generating synthetic data. The most common is to use some form of directed graphical model. In some cases, the structure of the model is designed to match the causal structure of a realistic system, either by manually specifying the structure or by learning it from empirical data. Large-scale simulators designed for other reasons can also be used. In some cases, simulators can be complex enough to generate data that is effectively equivalent to empirical data, though such simulations vary in quality.

Synthetic data is easy to collect, allows for straightforward comparison with ground truth, and facilitates systematic testing across a variety of data parameters. Its popularity is evident—84% (74) of surveyed papers used it in their evaluation, and 41% (30/74) of those used only synthetic data. However, using synthetic data for evaluation also has significant limitations. These include:

*Unquestioned assumptions*—Synthetic data tends to match the assumptions of the researcher running the study and any algorithms they have created. For example, a researcher developing an algorithm that outputs a DAG will be inclined to generate data from a DAG.

*Unknown influences*—Even the best data generators can only include the influences already known to researchers. Almost by definition, synthetic data generators cannot include any "unknown unknowns" that may influence the outputs of real-world systems. While latent variables can be added, they are still defined and created by the researcher, limiting the realism of the data.

*Lack of standardization*—Synthetic data is typically generated differently by each researcher, and this lack of standardization impedes comparison between studies.

*Researcher degrees-of-freedom*—Synthetic data is typically designed and parameterized by the researchers who created the algorithm being evaluated, giving them an enormous range of choices. Such high "researcher degrees-of-freedom" [Simmons et al., 2011] are a basic challenge to the validity of any study.

These factors significantly limit the external validity and realism of most synthetic data, making it insufficient as the sole source of data for evaluation. Synthetic data is not without value—it can be a powerful way to assess features of an algorithm and test its performance under different conditions. However, it typically falls short in providing insights into how the algorithm will perform on data from a real-world system.

## 3.2 Benefits of Empirical Data

Empirical data is almost always more difficult to collect than synthetic data, and information on the effects of interventions is typically also much more difficult to obtain. However, using empirical data has multiple benefits:

*Realistic complexity*—Empirical data typically has a distribution that is more complex than synthetic data. That distribution is subject to realistic latent factors and measurement error. This creates a learning task that is often significantly harder than synthetic data, but also more closely matches the challenges of real-world settings.

*Lower potential researcher bias*—Empirical data is typically not generated by the researcher who designed the algorithm being evaluated, and thus it is less subject to unintentional biases. In addition, individual data sets are often shared across the community, creating standardization and comparability across studies.

*Real-world demonstration*—The aim of research on algorithms for causal modeling is to have these algorithms used by others to learn causal models and reason about causal effects in real-world settings. Practitioners considering use of these methods may be legitimately skeptical about their effectiveness until they see successful demonstrations of accurate causal modeling on real-world data.

However, using empirical data poses challenges as well. Because it is generally not collected by the person using it, some features of the data may not be fully understood, hindering correct interpretation. Also, ground truth can be challenging to obtain, limiting evaluation to visual inspection or observational measures. This is unsatisfying at best and misleading at worst, since, when evaluating without ground truth, it can be easy to see meaning where none exists or to imagine explanations for many possible conflicting outputs. Despite these challenges, empirical data is still used frequently in practice; 67% (61) of surveyed papers use empirical data, and 28% (17/61) used only empirical data.

## 3.3 Sources of Empirical Data

Types of empirical data vary depending on the level of ground truth and the source of the ground truth. Purely observational data is the most readily available and is used most often. While this is rarely accompanied by full knowledge of the underlying structure, there are generally some dependencies that are known, either from common sense knowledge (such as temporal ordering) or from dependencies that have already been established by prior work. For a randomized controlled trial, the dependence between the measured treatment and outcome is generally taken as ground truth. The same is true for cases in which multiple potential outcomes can be recorded for each unit. This includes gene regulatory networks, flow cytometry analysis, and software systems, where essentially identical units can receive multiple treatments and thus produce multiple potential outcomes.

Because interventional measures and empirical data are used so infrequently, one might assume this is because such data sets are difficult to obtain. This is partially true—there are significantly more observational data sets available than interventional data sets. However, a growing community is producing data sets that provide interventional effects. We describe some of them here.

The cause-effect pairs challenge [Mooij et al., 2016] provides data that is empirical and, while interventional effects are not available, the direction of causality is known. The 2016 Atlantic Causal Inference Conference Competition and subsequent competitions [Dorie et al., 2019, Hahn et al., 2019] created semi-synthetic data sets, producing synthetic treatment and outcome functions using covariates from a real-world system. A similar approach was used by Shimoni et al. [2018] for the IBM Causal Inference Benchmarking Framework. Flow cytometry data, measuring protein signaling pathways, is another common choice for interventional data [Sachs et al., 2005]. Dixit et al. [2016] provide data on gene expression, collected using their proposed Perturb-Seq technique to perform gene deletion interventions. There has also been work in partially randomized experiments, where a population is split into randomized and observational groups, creating parallel datasets for evaluation [Shadish et al., 2008]. Other sources of interventional and empirical data include results of advertising campaigns [Sun et al., 2015] and clinical studies [McDonald et al., 1992], as well as multiple challenges organized for machine learning conferences [Guyon et al., 2008, 2010]. Domain specific simulations are another useful source of data. While technically synthetic, a sufficiently sophisticated simulation falls on a spectrum between purely synthetic and purely empirical data. They are often highly complex, are created by someone other than the researcher, and are created for a purpose other than evaluation, making them ideal for evaluation. One popular simulation that is used for evaluation is the DREAM in silico data sets, since multiple combinations of single-gene interventions can be performed on identical networks [Schaffter et al., 2011].

We also introduce an additional source of empirical data where interventions are possible: large-scale software systems. These systems have many desirable properties for the purposes of empirical evaluation: (1) They are pre-existing systems created by people other than the researchers for a purpose other than evaluating algorithms for causal modeling; (2) They produce non-deterministic experimental results due to latent variables and natural stochasticity; (3) System parameters provide natural treatment variables; and (4) Each experiment is recoverable, allowing the same experiment to be performed multiple times with different combinations of interventions. Three such data sets are discussed in more detail in Section 5 and in the Supplementary Material.[2]

## 3.4 How Different are the Results?

Readers may ask: In practice, what's the difference between using empirical data rather than synthetic data? If that difference is small, then the substantial extra work involved in evaluation with empirical data may not be worth the effort.

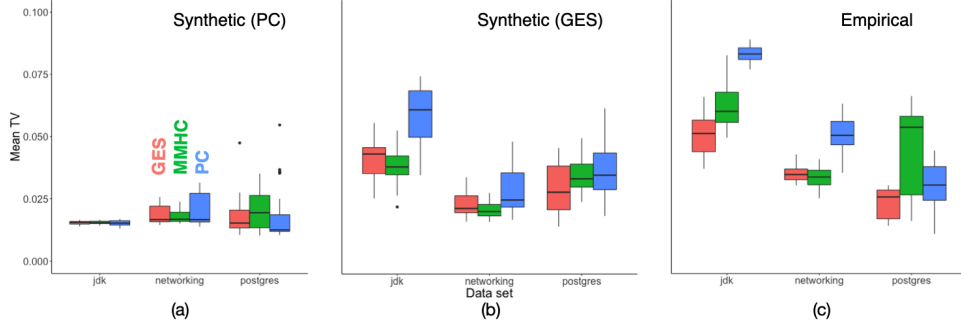

Figure 1: Comparison of TVD on empirical data and synthetic data derived from empirical data. **(a)** and **(b)**: synthetic data with structure obtained from PC or GES. **(c)**: TVD on empirical data.

To begin addressing this question, we conducted a series of experiments using interventional data from the software systems mentioned above. For these experiments, we used a common approach for generating somewhat realistic synthetic data. This approach uses an empirical data set to learn a causal model and then uses that model to generate synthetic data (and known ground truth effects) for model evaluation. While the final data set is synthetic, its structure may better approximate the empirical system, rather than being entirely defined by the researcher, lending it more credibility. We used this approach to generate synthetic data in the style of the three empirical data sets we generated from software systems. Since we now have both empirical and synthetic data, each with ground truth, we can use causal modeling algorithms to construct a model for both of these data sets and compare the conclusions we would draw from each.

The synthetic data used was created by first choosing an initial causal modeling algorithm to create a ground truth model from the empirical data. After learning a ground truth model with each of two algorithms that construct causal graphical models (PC and GES),[3] we generated synthetic data using the resulting models. We then evaluated the same three algorithms on both the synthetic and empirical data. Figure 1 shows how mean TVD varies for different causal modeling algorithms and different data sets. The results shown are the mean TVD when evaluating PC, GES, and MMHC on two types of synthetic data sets (using the model as ground truth) and on the empirical data (using the known interventional effects). There is significant variability between the two methods of generating the synthetic ground truth model from the empirical data (PC and GES), both in the mean TVD and in the relative ordering of the algorithms. Comparing the synthetic and empirical results, some relative orderings of the algorithms are the same (e.g., network), but other orderings are significantly different (e.g., Postgres). These results suggest that algorithm performance cannot be expected to match between synthetic and empirical data, even when the synthetic data is created in a way that would be expected to match aspects of the empirical data.

## 4   The Case for Interventional Measures

Many algorithms are currently evaluated based on their ability to learn causal structure. However, the actual desired task is almost never to model structure alone. In practice, estimating the magnitude of interventional effects is vitally important, and an algorithm that cannot distinguish between strong and weak effects is severely limited in scope. Despite this, the majority of current evaluations use observational or structural measures rather than measures of interventional effect.

### 4.1   Limitations of Observational Measures

Observational measures are widely used to evaluate algorithms for associational modeling, where the task of the algorithm is to discern statistical associations between two or more variables. In such applications, the primary focus is effectively modeling the magnitude and form of statistical dependence, rather than explicitly learning causal dependence. This highlights a severe and obvious limitation of observational measures:

*Non-causal*—Observational measures are, by definition, not causal. They measure the error of estimates of the outcome variable, but they do not measure that error under intervention. They provide a sense of how well an algorithm has learned statistical dependence, but not how well it has learned causal dependence. Despite this, observational measures are the *only* evaluation used in 23% (21/91) of papers surveyed.

## 4.2 Limitations of Structural Measures

Structural measures are easy to calculate, and they have a clear intuition. If an algorithm produces a causal structure and we know structural ground truth, it seems sensible to determine if the two structures match. This has led to the widespread adoption of structural measures: 55% (50) of surveyed papers used such measures, and 84% (42/50) of those used only structural measures. However, structural measures have several serious limitations:

*Requires known structure*—Calculating structural measures requires a full ground-truth graph structure, which is only rarely available for empirical data.

*Constrains research directions*—The prevalence of structural measures may constrain research to algorithms that can be evaluated with these measures. Algorithms that do not produce DAGs are less likely to be developed or favorably reviewed. Since structural measures can only be used by algorithms that produce a directed graphical model as output, they implicitly assume that directed graphical models are capable of accurately representing any causal process being modeled, an unlikely assumption.

*Oblivious to magnitude and type of dependence*—Structural measures, by design, do not account for different magnitudes of dependence, so an error in an edge with a strong effect incurs the same penalty as an error in an edge with a very weak effect. In addition, structural measures are only able to measure *which* variables in a causal model change as the result of an intervention. In many cases, it is also necessary to determine *how much* or *in what way* a given target quantity will change with respect to an intervention.

*Oblivious to likely treatments and outcomes*—In most cases, structural measures do not consider where an edge is located in the overall structure of the DAG, so an edge with many downstream effects is treated the same as a less central edge.

## 4.3 Benefits of Interventional Measures

In contrast to observational and structural measures, interventional measures have strong advantages:

*Correspondence to actual use*—Interventional measures evaluate how well the model estimates interventional effects, which aligns more closely with the eventual use of nearly all causal models. For example, a directed acyclic graph is not the ultimate artifact of interest for most applications— DAGs are simply a representation that facilitates estimation of interventional effects [Spirtes et al., 2000, Pearl, 2009]. Thus, it seems natural to define an evaluation measure in terms of interventional effects rather than graphical structure.

*Weighting of different errors*—While most structural measures penalize each edge misorientation equally, interventional measures penalize misorientation errors proportionally to their effect on the estimation of interventional effect.

## 4.4 How Different are the Results?

Interventional measures are intended to capture something different than structural measures, but they are ultimately affected by the structure of the learned model, and we would expect structural errors to lead to errors in interventional effect estimates. Of course, interventional and structural measures are equal when structure and parameterizations are perfect, but they can differ significantly when the learned structure is only approximately correct (which is almost always the case). To assess the extent to which interventional measures capture different information than structural measures in such cases, we ran experiments using synthetic data. This allowed us to produce data where we could calculate both structural measures and interventional measures, since we had the full parameterized ground truth model to compare against.

For these experiments, we produced data from random DAG structures with conditional probability models drawn from a Dirichlet distribution. We generated 5000 instances, applied a causal modeling algorithm, and calculated various evaluation measures. Figure 2 shows the results for GES. SHD and SID are clearly strongly correlated, suggesting that both structural measures ultimately produce similar quality measures of the algorithm. However, SHD and TVD are only very weakly correlated, with many models scoring highly with one measure and poorly with the

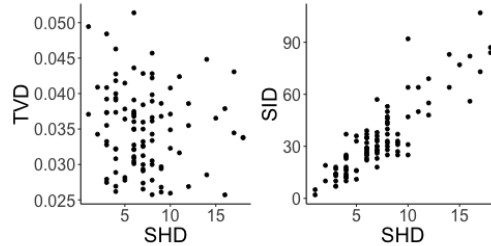

Figure 2: Structural and interventional measures compared on synthetic data with GES.

other. At least in this case, the interventional measure (TVD) appears to capture substantially different information than that of a structural measure (SHD). Results for PC and MMHC are reported in the Supplementary Material.

## 5 Example of an Evaluation

To further explain what we mean by empirical data and interventional measures, we describe one example of this type of evaluation, shown schematically in Figure 3. This example demonstrates one way that an evaluation with empirical data and interventional measures could be performed, though many other techniques are possible, depending on the algorithm, data source, evaluation measure, and the research question under consideration. In our example, we evaluate the PC algorithm [Spirtes et al., 2000], Greedy Equivalence Search (GES) [Chickering, 2003], and MMHC [Tsamardinos et al., 2006] by measuring *total variation distance* (an interventional measure defined later) on a data set produced by experimentation with a large-scale software system.

An obvious way to evaluate how well an algorithm can learn causal models from real-world data is to compare the model's estimates to empirical data drawn from a system in which we can perform multiple interventions on the same units, giving us full interventional data in which we can assess every potential outcome for each unit. Large-scale software systems allow for this type of intervention because they let us run the same experiments multiple times under different conditions (e.g., different settings of key system parameters). An example of this is a Postgres database, where we can run the same queries with different settings of key configuration parameters. In this context, each query corresponds to a unit, a set of configuration parameters correspond to treatment, and variables such as runtime correspond to outcomes. Details about this data can be found in the Supplementary Material.

Many algorithms for causal modeling are designed to run on observational data, in which only a single, non-randomized treatment assignment is observed for each unit. In the absence of an observational data set that matches our interventional data, we can create an observational-style data set by sub-sampling the full interventional data in a non-random manner. To do this, we select a single treatment assignment for each query. Selecting treatment at random is equivalent to a randomized controlled trial. In most observational contexts, however, treatment assignment would be based on covariates of the units. For example, a database administrator might choose the configuration parameters based on features of each query. We use a similar process to create observational data by using a measured covariate of the query to probabilistically assign treatment.

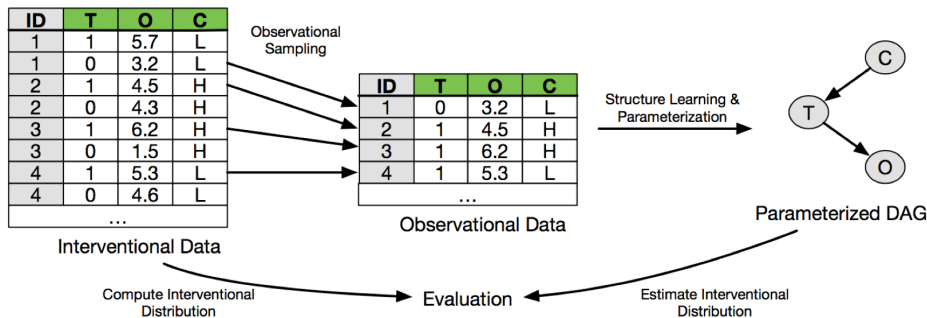

Figure 3: A diagram of one approach to evaluating a causal modeling algorithm

Given such an observational data set, we can apply a causal modeling algorithm and learn a causal model. A fully parameterized model can produce an estimated interventional distribution $\hat{P}$ by applying the *do*-calculus [Galles and Pearl, 1995]. Under this framework, causal quantities take the form of probability queries with *do* operators, for instance $P(O|do(T = 1))$. We can also estimate the actual interventional distribution $P = P(O = o|do(T = t))$ for any outcome $o$ and treatment $t$, because we can measure the effects of both values of treatment for each query in our data set.

We then can use an interventional measure to compare the true interventional distribution $P$ to the estimated distribution $\hat{P}$. One example of an interventional measure is total variation distance (TVD) [Lin, 1991], which measures the distance between two probability distributions. For discrete outcomes $O$, the quality of an estimated interventional distribution relative to a known distribution under TVD is straightforward to compute:

$$TV_{P,\hat{P},T=t}(O) = \frac{1}{2} \sum_{o \in \Omega(O)} \big| P\left(O = o|do(T = t)\right) - \hat{P}\left(O = o|do(T = t)\right) \big|,$$

where $\Omega(O)$ is the domain of $O$. This gives us a numerical measure of how well the estimated interventional estimates match the ground truth. A single TVD value is computed for each causal effect, which can then be aggregated for comparison. Results of this evaluation on the software data is shown in Figure 1c. For these datasets, we can conclude that GES has the best overall performance.

# 6    Conclusion

Evaluation is a key mechanism that determines how algorithms are viewed within the community, what research directions are pursued next, and whether our research has broader impacts outside the community. Our current evaluation techniques aim too low, and they fail to evaluate the full range of questions that our research goals imply.

We are not the first to point out the need for more robust evaluation techniques. Some of the datasets we discuss were created in response to recognition that better evaluation was necessary [Dorie et al., 2019, Shimoni et al., 2018, Mooij et al., 2016]. In addition, prior work has examined the importance of testing the generalizability of causal inferences drawn from observational data [Zhao et al., 2019, Keane and Wolpin, 2007] and comparing causal effects drawn from observational and experimental data [Cook et al., 2008, Eckles et al., 2016, Eckles and Bakshy, 2017, Gordon et al., 2019]. However, despite this, as our survey shows, empirical evaluation with interventional measures is rarely used by computer science researchers.

We acknowledge that, while the evaluation techniques we advocate are applicable to a wide range of algorithms, data sets may not be available for every task. The diverse tasks of causal modeling algorithms make it difficult to recommend a single data set and evaluation measure to evaluate every algorithm. However, the data sets and measures that are most commonly used are largely insufficient. The community would benefit if more data sets with interventional effects were created and made available for public use, allowing for a breadth of evaluation options.

We do not advocate abandoning synthetic data and structural measures. Both have many uses for evaluating algorithm performance and can be indispensable scientific tools. However, they are insufficient on their own. Instead, they should be viewed as a first step in evaluation. If we want causal modeling algorithms to be adopted outside our research community, we need demonstrations of their utility outside of a laboratory setting. If we do not evaluate on empirical data, we cannot be certain our algorithms will perform well on real-world data, and if we do not evaluate with interventional measures, we cannot be certain that the causal effects our algorithms infer will translate to actual, substantial causal effects in practice. Expanding our routine evaluations will substantially improve the credibility and comparability of results, the external validity and trustworthiness of algorithms, and the efficiency with which we conduct our research.

**Acknowledgments**

This material is based upon work supported by the United States Air Force under Contract No, FA8750-17-C-0120. Any opinions, findings and conclusions or recommendations expressed in this material are those of the author(s) and do not necessarily reflect the views of the United States Air Force.

## Footnotes

[1]When reporting survey results, we follow each percentage with a parenthesized number representing the raw count. The denominator for percentages is 91, except where otherwise noted.

[2]These data sets are available for download at http://kdl.cs.umass.edu/data.

[3]We reach similar conclusions based on the results for MMHC, which are reported in the Supplementary Material.

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
