[Supplementary Material · NeurIPS-causal-eval-supplement.pdf]

# Supplementary Material

## 1    Additional Details on Software Data

We introduce a source of empirical data where interventions are possible: large-scale software systems. We performed experiments on three large computational systems: Postgres, the Java Development Kit, and HTTP processing. These systems have many desirable properties for the purposes of empirical evaluation: (1) They are pre-existing systems created by people other than the researchers for a purpose other than evaluating algorithms for causal discovery; (2) They produce non-deterministic experimental results due to latent variables and natural stochasticity; (3) System parameters provide natural treatment variables; and (4) Each experiment is recoverable, allowing the same experiment to be performed multiple times with different combinations of interventions.

Within each computational system, we measure three classes of variables: outcomes, treatments, and subject covariates. Here, outcomes are measurements of the result of a computational process, treatments correspond to system configurations and are selected such that they could plausibly induce changes in outcomes, and subject covariates logically exist prior to treatment and are invariant with respect to treatment. Using these variables, we can apply all combinations of treatments to all subjects, and we can use these results to estimate actual interventional distributions for the effects of each treatment variable on each outcome variable. We can also then sub-sample these experimental data sets in a manner which simulates observational bias to produce observational-style data sets, allowing us to evaluate an algorithm's performance on pseudo-observational data and evaluate it using actual interventional effects. These data sets will be made available after publication.

We had a number of goals in mind when gathering data from our real domains:

- **Causal Sufficiency:** The algorithms we studied require that no pair of variables in the model are both caused by a latent variable. We can guarantee this is true for pairs of treatments and outcomes (since treatments have no parents in the original data set), but needed to employ domain knowledge to limit sources of causal sufficiency violations with regard to other pairs of variables.
- **Acyclicity:** Each of the systems can be described by a "single-shot" computational process which starts and finishes without the possibility for feedback.
- **Instance Independence:** We took efforts to ensure that each execution of the computational process was independent of previous executions. In most cases, this required clearing caches and resetting other aspects of system state.
- **Plausible Dependence:** We selected variables that we believed would be causally related.

Each domain is characterized by three classes of variables: subject covariates, treatments, and outcomes. Under the factorial experiment design, outcomes were measured for every combination of subjects and treatments. This yields a data set with many records for the same subject, as in the example in Table 1. To permit greater opportunities for observational sampling, we performed multiple trials of each factorial experiment. Given the difficulty associated with modeling highly complicated outcomes such as runtime, we employed a normalization scheme for each data set, dividing outcome values by a "baseline" value—the median control-case outcome value. Thus, we ultimately recorded outcomes which represent a deviation from this baseline. In this regard, our experimental results resemble a within-subjects design Greenwald [1976], although without many of the pitfalls that plague experiments on humans, such as non-independence of outcome measurements. In the original data

| Subject ID | Covariate | Treatment | Outcome |
|:---:|:---:|:---:|:---:|
| 1 | A | 0 | 1.33 |
| 1 | A | 1 | 0.96 |
| 2 | B | 0 | 1.89 |
| 2 | B | 1 | 0.54 |
| 3 | A | 0 | 1.02 |
| 3 | A | 1 | 0.99 |
| 4 | A | 0 | 1.35 |
| 4 | A | 1 | 1.12 |

Table 1: An Example of a Factorial Experiment with Four Subjects and a Binary Treatment

Figure 1: Consistent Model for the JDK Domain

from each domain, subject covariates are either discrete, continuous, or binary; treatments are binary; and outcomes are continuous. We converted each of the variables to a discrete representation to make parametrization and inference more robust.

## 1.1 Java Development Kit

Our experiments on the Java Development Kit (version `1.7.0_60`) used 2,500 Java projects obtained from GitHub as the subjects under study. We retrieved only projects which use the Maven build tool to facilitate automated compilation and execution. Additionally, we constrained our search to include only projects which had unit tests. This may introduce selection bias in our data collection processes, but this is acceptable. It is not important that our conclusions generalize to some population of computational systems, only that there are causal dependencies which hold on the sub-population under investigation. Of those, 473 compiled and ran without intervention. This group yielded a total of 7,568 subject-treatment combinations. For each combination, we compile and execute the unit tests of the Java project. In order to obtain full state recovery between each trial, any compiled project files were cleared between executions. Thirty-five CPU days were required to collect this data using several Amazon EC2 instances.

### 1.1.1 Treatments

- **Aggressive Compiler Optimization**: Disabling this option (enabled by default) prevents some compiler optimizations from running, potentially slowing down execution time but perhaps reducing compilation time. This option is disabled with the `javac` option `-XX:+AggressiveOpts`.

- **Emission of Debugging Symbols**: Debugging symbols are used to provide a map through the compiled source code that can be used for interactive debugging and diagnostics. Inclusion of these symbols may require some time during the compilation phase, increase the size of the compiled program, and could possibly impact runtime. This corresponds to the `-g` flag of `javac`.

- **Garbage Collection Methodology**: The Java Development Kit supports several garbage collection schemes. Two were considered: parallel and serial. These schemes are activated with the `-XX:-UseParallelGC` or `-XX:-UseSerialGC` arguments.
- **Code Obfuscation**: Several third-party tools are capable of obfuscating compiled code, making reverse-engineering difficult. This process could also affect the size of the compiled project files. The yGuard[1] tool was used for this purpose.

### 1.1.2 Outcomes

- **Number of Bytecode Instructions**: Before execution, Java code is compiled to an intermediate language referred to as bytecode. We measured the number of atomic instructions, or operations, in this compiled code to form this outcome using a custom-built bytecode analysis tool based on Javassist[2].
- **Total Unit Test Time**: Each project we gathered contains one or more unit tests. To capture the runtime of the full unit test workload, we computed the sum of runtimes of all unit tests for a given project.
- **Allocated Bytes**: The Java Virtual Machine supports a profiling option (`-agentlib:hprof=heap=sites`) which can be used to track heap statistics throughout a program's execution. We utilized this feature to obtain the total number of bytes allocated during unit test execution.
- **Compiled Code Size**: Java programs are often packaged in an format known as a JAR (Java ARchive). To characterize the size of the compiled code, we recorded the size in bytes of the associated JAR file.
- **Compilation Time**: In order to execute unit tests, the entire project needs to be compiled. This outcome represents the time used to convert all source files to their bytecode equivalents.

### 1.1.3 Subject Covariates

All subject covariates were obtained using the JavaNCSS tool[3].

- **# NCSS (non-comment source statements) in Project Source**: This covariate is highly predictive of compiled code size. Conceivably, in observational settings, large projects could also be associated with more liberal use of advanced compilation settings and tools, such as a code obfuscator.
- **# NCSS, Functions, and Classes in Unit Test Source**: These covariates are somewhat representative of the unit test workload. Projects with many lengthy unit tests may also have longer total unit test runtime.
- **# "Javadoc" comments in Unit Test Source**: This covariate could be indicative of code quality. Well-commented code is perhaps more likely to be found in high-quality projects. This code may be more likely to be used in production environments, and thus could be less likely to be observed with debugging symbols. This feature is used in the treatment-biasing procedure for construction of observational data sets.

## 1.2 Postgres

Consistent with a data warehousing scenario, we employ a fixed database for our Postgres (version 9.2.2) experiments: a sample of the data from Stack Overflow, drawn from the Stack Exchange Data Explorer[4]. The data explorer also houses many user-generated queries. We collected 29,375 of the most popular queries to use as subjects for this study. Stack Exchange's data warehouse uses Microsoft SQL Server, which does not completely overlap with Postgres in supported features and syntax. Some queries use only ANSI-compliant syntax and run successfully on either SQL Server or Postgres. To obtain as large a set of subjects as possible, we employed a semantics-preserving

Figure 2: Consistent Model for the Postgres Domain

query rewriting scheme to adapt queries into Postgres-compliant syntax wherever possible. This yielded a set of 11,252 user-generated queries which executed successfully within Postgres for a total of 90,016 subject-treatment combinations. In order to recover system state between trials, the shared memory setting (specifying how much main memory Postgres can use for caching) was set to 128 kilobytes, limiting caching significantly. Any queries which required more than 30 seconds to execute were marked as "failures" in order to prevent long-running queries from holding up other queries, which typically required one second to execute. As with the JDK data set, this may induce sampling bias, but we are not aiming for our experimental findings to generalize to the broader population of database queries.

### 1.2.1 Treatments

- **Indexing**: A common administration task is to identify indices that can be used to accelerate lookup of commonly-referenced columns with a particular value or falling within a range. For our experiments, we employed two indexing settings: no indexing, and indexing on primary key/foreign key fields. Domain knowledge suggests that that the latter approach would dramatically reduce runtime of some queries. In all cases, the default B-tree index was employed.

- **Page Cost Estimates**: In order to determine if an index should be used, the database employs estimates of the relative cost of sequentially accessing disk pages and randomly accessing disk pages. We utilized two extremes for this setting: one scheme in which random page access is estimated to be fast, relative to the sequential page access, and one scheme in which the opposite relation holds. The corresponding database settings we adjusted were `random_page_cost` and `seq_page_cost`.

- **Working Memory Allocation**: The database engine can make use of fast random-access memory, if available, to store intermediate query results. The amount of working memory that is allocated to the system can be controlled with a configuration option. For our investigation, we employed a low-memory setting and a high-memory setting, with background knowledge suggesting that the latter would result in faster-executing queries. This treatment was instrumented with the `work_mem` and `temp_buffers` options.

### 1.2.2 Outcomes

- **Blocks Read from Shared and Temporary Memory**: These two outcomes identify the number of blocks, or memory regions, that were read during query execution. Shared memory is persistent (disk) and is accessed during normal table-retrieval procedures. Temporary memory is volatile (main memory) and is used for staging ordering or joining operations.

Figure 3: Consistent Model for the HTTP Domain

- **Blocks Hit in Shared Memory Cache**: This outcome represents the number of memory reads that were to be performed against shared memory, but were identified instead in a main memory cache.

- **Runtime**: The total time to execute the query.

### 1.2.3 Subject Covariates

- **Year of Query Creation**: The year that the query was entered on the Stack Exchange data explorer.

- **Number of Referenced Tables**: The number of distinct tables that are referenced in the query.

- **Total Number of Rows in Referenced Tables**: The sum of cardinalities of tables referenced in the query.

- **Number of Join Operators**: The number of join operators employed in the query, requiring merging data from two tables.

- **Number of Grouping Operators**: The number of grouping operators employed in the query, requiring reduction and possibly summarization of the data.

- **Number of Other Queries Created by the Same User**: The total number of queries that the Stack Exchange user has created.

- **Length of the Query in Characters**: The length of the query after application of relevant rewrite rules.

- **Number of Rows Retrieved**: The number of rows that are returned by the query. Logically, this value exists prior to application of any treatment and is invariant with respect to treatment (since the database is fixed), even though we can only measure it after query execution.

### 1.3 Hypertext Transfer Protocol

For our experiment on HTTP & networking infrastructure, we used requests to specific web sites as subjects. We identified a number of target sites through a breadth-first web crawl initiated at dmoz.org. We ended the crawl after retrieving 5,472 sites. For 4,350 of those sites, we were able to issue successful web requests with all combinations treatments, yielding 34,800 subject-treatment combinations. We employed numerous techniques to ensure that content would not be cached, which could induce carryover across treatment regimes.

### 1.3.1 Treatments

- **Use of a Mobile User Agent**: Web browsers supply a *user agent* to identify themselves to the web servers that they request pages from. Some sites have different versions for mobile applications. We artificially adjusted the user agent from a standard user agent to a mobile user agent to explore this phenomena. This is accomplished with the HTTP `User-Agent` header.

- **Proxy Server**: Web requests can be routed through a *proxy*, a server which issues web requests on behalf of a client. The additional time required to route the request to and from the proxy server can increase the elapsed time of the request. Our experiments were executed with Amazon EC2. Our "client" computers were making web requests from the east cost of the United States, and a proxy server was set up on the west coast.

- **Compression**: Applications can use the HTTP protocol to request that content be delivered with or without compression, possibly reducing the cross-network transmission time. In one compression configuration, the client requests `identity` compression, indicating that the content should be transmitted at face value. In another compression scheme, the client requests `gzip`, a common and effective scheme for HTTP content compression.

### 1.3.2 Outcomes

- **# of HTML Attributes and Tags**: These two outcomes describe the logical structure of the page. They may vary with respect to "mobile user agent".

- **Elapsed Time**: The time between issuance of the request and receipt of a response. This could be affected by network characteristics, which are determined in part by the time at which the request is issued and whether a proxy server is employed. Requests containing smaller payloads (influenced by compression) may also be faster to service.

- **Decompressed and Raw Content Length**: Two outcomes representing the size of a web page before and after content decompression, if applicable.

### 1.3.3 Subject Covariates

Only one subject covariate was identified for the HTTP domain, the web server reported via the `Server` header. This variable was coarsened into a version with 7 levels: Apache/2, Other Apache, Microsoft-IIS, nginx, Other, and Unknown.

## 2 Identifying Consistent DAGs

To identify DAGs that can consistently estimate the all interventional distributions $P(O|do(T))$, we need to ensure that (1) the parent set of $T$ is a valid adjustment set with respect to $O$, and (2) if $T$ has a causal effect on $O$, there is a chain connecting $T$ and $O$ in the DAG model. The first condition is straightforward to satisfy since we know the only parent of any treatment to be the covariate used to introduce observational bias. The second condition requires identification of which pairs of treatments and outcomes are causally related. These *d*-connection properties were identified for each domain using the full interventional data set using the Friedman test for blocked difference in means, allowing for correction of subject variability Friedman [1937]. An edge was introduced between any causally related pair to satisfy condition (2). Then, ground truth interventional distributions $P(O|do(T = t))$ were produced by applying the do-Calculus model adjustment rules, and answering probability queries $P(P|T = t)$ on the resulting model using belief propagation.

## 3 Pseudo-Observational Configurations

We can transform the factorial experiments on our real domains into pseudo-observational data by sub-sampling the experimental data in a way that is correlated with a "subject covariate". This mirrors the process of treatment self-selection common to observational data. This transformation is outlined in Algorithm 1.

**Input:** Interventional data set $I$, biasing strength $\beta \geq 0$, biasing covariate $C$
**Output:** Observationally biased data set $O$, $|O| = nd$
$l \leftarrow$ The number of distinct values of $C$
**foreach** *Subject* $e \in I$ **do**
    Let $C_e \in \{1..l\}$ represent the $C$ value of subject $e$
    $Assign \leftarrow \{\}$
    **foreach** *Treatment* $T_j$ **do**
        $s_{ej} \leftarrow \begin{cases} 1 & \text{if } C_e \times j \text{ is even} \\ -1 & \text{if } C_e \times j \text{ is odd} \end{cases}$
        $p \leftarrow \text{logit}^{-1}(s_{ej}\beta)$
        $t_j \leftarrow \text{Bernoulli}(p)$
        $Assign \leftarrow Assign \cup \{T_j = t_j\}$
    **end**
    $M \leftarrow$ Record in $I$ corresponding to $(e, Assign)$
    $O \leftarrow O \cup M$
**end**

**Algorithm 1:** Logistic Sampling of Observational Data

## 4   Limitations of Empirical Data

In the paper, we discuss popular sources of empirical data that is suitable for evaluation. These data sets differ significantly in many ways, including level of realism and data quality, and they each have different benefits and limitations.

The cause-effect pairs challenge [Mooij et al., 2016] provides observational data on pairs of variables where the direction of causality is known from domain knowledge. This data set is useful for evaluating bivariate orientation algorithms, but the lack of any additional measured covariates limits its utility for evaluating multivariate structure learning algorithms.

The 2016 Atlantic Causal Inference Conference Competition data [Dorie et al., 2019] and the IBM Causal Inference Benchmarking Framework Shimoni et al. [2018] use covariates taken from a real-world data set, allowing for potentially complicated interactions between them. Treatment and outcome functions were then generated synthetically, using a variety of data generating processes to allow for the construction of many data sets with different features. This allows algorithms to be tested on many data sets, providing a more robust evaluation. However, the need to construct synthetic treatment and outcome functions limits the level of realism.

The software data we collected contains measurements of covariates, treatments, and outcomes from three real-world systems. While the treatment function is generated synthetically, the outcome function is not, lending the ground truth causal effects from treatment to outcome a high degree of realism. However, as with the above ACIC and IBM data sets, the treatment function still needs to be synthetically defined.

The flow cytometry data provided by Sachs et al. [2005] contains measurements of protein signaling pathways, where multiple activating and inhibitory interventions were performed. However, the ground truth is not clearly obtainable and most analysis using this dataset relies on structural measures.

Partially randomized experiments, where a population is split into randomized and an observational groups, are another useful source of empirical data [Shadish et al., 2008]. The collection of randomized data drawn from the same base population as observational data creates a convenient ground truth for causal effect estimation. However, due the nature of these experiments, they require careful experimental design to make sure the populations are equvalient and the treatments are correctly assigned and measured.

The DREAM in silico data sets [Schaffter et al., 2011] are taken from a sophisticated simulation derived from multiple known gene regulatory network structures, which, while non-empirical, is intended to be complex enough to approximate empirical data. However, realism is limited due to the use of a simulator.

Figure 4: Relative Performance of Causal Discovery Algorithms on Synthetic Data Sets

## 5 Additional Experiments

In the paper, we provided experiments that demonstrate that TVD and structural measures provide different information and that information is relevant for over and under specification. To expand on these results, we performed an additional experiment to evaluate if different types of measures would lead to different conclusions about the relative performance of causal modeling algorithms. Figure 4 shows results on synthetic data that demonstrate that TVD does, inf act, imply a very different ordering of the relative performance of different learning algorithms than that implied by SHD and SID. We began by constructing 30 random DAGs with 14 variables and $E[N] = 2$. We generated parameters on those DAGs using each of the synthetic data techniques and sampled 5,000 data points from each DAG. Then, we applied PC, MMHC, and GES to the resulting data sets and measured the SID, SHD, and sum of pairwise total variations. As shown in Figure 4, some of the findings that would be reached with SID and SHD are not supported by a TV evaluation. The structural measures suggest that MMHC outperforms PC on the Dirichlet domain. However, the performance of the two algorithms is statistically indistinguishable as measured by TV. When measured with SID or SHD, GES does not outperform either MMHC or PC. However, GES is consistently the best performing algorithm in terms of interventional distribution accuracy.

Experiments in the paper demonstrate that TVD can, at least in some cases, provide information that structural measures cannot. However, that does not mean that the additional information is

Table 2: Metric Comparison on Real Domains with Over-specification and Under-specification

| Domain | Subjects | Model Type | SID: Min, Median, Max | | | SHD: Min, Median, Max | | | TVD: Min, Median, Max | | |
|---|---|---|---|---|---|---|---|---|---|---|---|
| JDK | 473 | Over-specify | 0 | 0 | 0 | 1 | 3 | 3 | 0.04 | 0.17 | 0.21 |
| | | Under-specify | 4 | 5 | 9 | 2 | 2 | 4 | 0.22 | 0.41 | 0.58 |
| Postgres | 5,000 | Over-specify | 0 | 0 | 0 | 0 | 1 | 2 | 0.00 | 0.06 | 0.09 |
| | | Under-specify | 4 | 6 | 8 | 3 | 4 | 5 | 0.17 | 0.35 | 0.61 |
| HTTP | 2,599 | Over-specify | 0 | 0 | 0 | 1 | 2 | 4 | 0.06 | 0.06 | 0.09 |
| | | Under-specify | 2 | 6 | 10 | 1 | 3 | 4 | 0.22 | 0.25 | 0.30 |

useful. To address this concern, we sought to measure how TVD responds to specific types of errors in learned structure. Specifically, we evaluate the effects of over-specification (extraneous edges) and under-specification (omitted edges) on model performance. We used our three empirical data sets drawn from large-scale computational systems (JDK, Postgres, and HTTP) to perform this analysis. From the original exhaustive experiments, we can identify which treatment-outcome pairs are causally related. We construct a partial DAG, consisting only of edges between treatment and outcome, by introducing an edge between each pair of causally related treatment and outcome. Then, a pseudo-observational data set can be constructed by sub-sampling treatment assignments according to a biasing covariate (details in Supplemental Materials). The resulting DAG model (illustrated for the JDK data set in Figure 1) consistently estimates distributions $P(O|do(T = t))$ for all treatment-outcome pairs.

We altered the consistent models of each data set to induce over-specification and under-specification. To quantify the effects of over-specification, we produced models in which one of the treatment variables had a directed edge into every outcome, regardless of the causal relationships in the true model. To quantify the effects of under-specification, we produced models in which one of the treatment variables had no outgoing edges. This process was repeated for each of our three domains and each treatment variable within that domain. For each model, a sum of pairwise total variations was computed as $\sum_{T,O} TV_{P,\hat{P},T=1}(O)$, where $P$ represents the reference distribution given by the consistent model (as in Figure 1) and $\hat{P}$ represents the distribution induced by the altered model. A comparison of TVD, SHD, and SID on these experiments is shown in Table 2.

Two properties are apparent. First, over-specification is penalized differently by different evaluation measures. For small data sets, such as the JDK domain, over-specified models have zero SID but significant TVD values due to loss of statistical efficiency. Second, penalizing over-specification and under-specification with equal cost, as in SHD, is inconsistent with interventional distribution quality. In these domains, model under-specification has 2-5x the distributional impact of under-specification as measured by total variation.

## 6 Additional Details on Presented Experiments

Figures 5 and 6 show the results of comparing synthetic and interventional measures on synthetic data for both MMHC and PC. (results for GES were presented in the paper) Interestingly, while the correlation between SID and SHD is relatively consistent for all three structure learning algorithms, the correlation between TVD and SHD varies substantially, from seemingly completely uncorrelated (GES) to very clearly correlated (PC). This suggests that, in some cases, structural measures can provide a decent proxy for interventional measures. However, it is unlikely that the researcher knows this to be the case ahead of time, and the comparative difference in TVD between the three algorithms suggests the value of using TVD when comparing multiple causal learning algorithms.

We also provide additional results for experiments discussed in the paper that created synthetic data sets by learning their structure from empirical data. While we reported results using GES and PC, here we show results for MMHC. Figure 7 shows the performance of three learning algorithms (GES, MMHC, and PC). MMHC was used to infer a causal model from empirical data, and that model was then used to generate the synthetic data. Compared with the results in the paper, the relative performance of different algorithms looks somewhat similar to the results using GES, though there are some differences (e.g., PC is clearly the worst on all data sets in Figure 7, while this is not the case for GES in Figure 1 in the paper).

Sample sizes for some of the software system data sets are small, so in Figure 7 and Figure 1 in the paper, we report results as distributions over 30 trials for each algorithm and data set.

Figure 5: Structural and Interventional Measures Compared on Synthetic Data with MMHC.

Figure 6: Structural and Interventional Measures Compared on Synthetic Data with PC.

Figure 7: Results for MMHC for the Experiments Described in the Paper, using Synthetic Data that has been Created to Look like Empirical Data

## Footnotes

[1] http://www.yworks.com/en/products_yguard_about.html

[2] http://www.csg.ci.i.u-tokyo.ac.jp/ chiba/javassist/

[3] http://javancss.codehaus.org/

[4] http://data.stackexchange.com/