[Reviews · NeurIPS 2019]

Reviewer 1



Update: I thank the authors for addressing my comments. I increased my recommendation from 6 to 7 as a result. Originality: Perhaps the biggest problem with the paper. The point it is making is a good one –– one that needs to be heeded by researchers in the field. But it is not a *new* contribution. The paper seems to ignore previous work on the topic in the statistics literature (Cook, Shadish and Wong 2008; Zhao, Keele and Small 2018... just to name a few). There are a number of references in the statistics literature that make a similar point, with more details. Clarity: the paper is well generally clear and well written. Quality: the paper makes good points, but has a somewhat limited scope. In particular, it does not provide any guidance for how to conduct a convincing empirical evaluation. This limits the potential impact of the paper, in my opinion. It would have been great if the authors had dedicated a couple of pages to reviewing some of the methods for empirical evaluation, or case studies of empirical evaluation done right.

Reviewer 2



Although the paper is a good attempt at this space, and the messages should be echoed wide in the community, the paper could benefit from various improvements. Specifically, I am unsure if some of the performed experiments are supportive of the claims made in the paper. Details are as follows: Line 79: Authors discuss evaluating interventional distribution. But if the structure learning part is correct, then the learned distribution will also be correct as long as the parameterization is known or for discrete variables. Am I missing a point here? After reading the rest, I guess authors are concerned about approximately learning the structure, and then depending on whether strong or weak edges are omitted can be determined by such an evaluation. It may help expand this discussion here a bit too. Can you elaborate a bit more on untested influences in line 176. Line 245: The data proposed in the machine learning challenges mentioned here is already used in the cause-effect pair dataset of Mooij et al. Section 4.4: Please explain this experiment of generating synthetic data on the learned network in more detail: How many samples were in the real data, how many samples did you generate synthetically? The mentioned algorithms can perform poorly if the number of samples are small, which is a different problem than using synthetic data. " Structural measures also implicitly assume that DAGs are capable of accurately representing any causal process being modeled, an unlikely assumption" This issue is much more complicated than authors imply. Once we remove the assumption that the underlying graph is acyclic, the modeling changes drastically. So, if an algorithm that is based on a set of well-defined assumptions including the assumption that the underlying graph is acyclic, outputs a cyclic graph it is a clear error and should be avoided. It is a different point to encourage assuming cyclic models and developing algorithms for that, but that is at the modeling phase, much before evaluation. Please elaborate this part as the quoted sentence can be misleading, diminishing the significant difference in modeling cyclic vs. acyclic systems. The TVD vs. structural distance plot is interesting. Is the TVD calculated only on the observational distribution. AFTER REBUTTAL: I would like to thank the authors for their detailed response. Although it clarified many points, I still believe the reason for seeing multiple outcomes from the causal inference algorithm is probably simply using insufficient number of samples, rather than synthetic vs. real data. I hope authors investigate this point better.

Reviewer 3



This clearly written and highly novel paper describes a critical gap in the causal inference literature. While inference methods have advanced, our evaluation techniques have not. As the authors show, this means that our ability to predict which methods will translate successfully to practice is limited. The paper contains a thorough survey of inference methods and evaluations, which i have not seen before. This is a valuable contribution to the literature. While the paper is not perfect (see improvements section), I believe the significant novelty and potential impact on the community outweigh these weaknesses and that it is a significant contribution. Figure 2 is especially striking. Questions: -The authors discuss the significant limitations of synthetic data. However, only simulations using the target structures (e.g. DAG) seem to be considered. What about using domain specific simulation systems? These are totally independent of the methodological assumptions/approaches. Do you believe the results would be closer to interventional measures? I appreciated the responses to the reviews, and maintained my high score as the weaknesses pointed out by the other reviewers will be addressed in revision.

[Author Response · NeurIPS 2019]

We thank all three reviewers for their very thoughtful and detailed comments, with which we largely agree. Some
requested additions (e.g., details about experiments and interventional datasets, an example of an empirical evaluation)
were present in previous versions of the paper and were removed for space. We will add these sections to any revised
version, either in the main body of the paper or the Supplemental Material. More details appear below.

**Additional citations (R1)**  We thank Reviewer 1 for the highly relevant pointers to the statistics literature—these will
be cited and discussed in any revised version. Within-study comparisons provide a great source of data that is
both experimental and observational and should be discussed as a source of interventional data.

**Originality (R1)**  Some of the points discussed in this paper have been touched on in previous work in statistics and
quantitative social science (e.g., the citations mentioned by Reviewer 1). However, this work is not widely
known in the machine learning community (the primary audience for NeurIPS), and this work often concerns
a highly specific evaluation context (i.e., single-treatment/single-outcome, average treatment-effect, etc.). As
seen by our survey, only a small fraction of the ML community performs evaluations that are both empirical
and interventional, and there are very few standard data sets in the field that allow for this. We believe that
further evidence, targeted at researchers within computer science and addressing the approaches and concerns
of that community, is necessary to promote the wider adoption of more principled evaluation methods.

**Evaluation guidance (R1)**  Reviewer 1 notes that the paper provides little guidance on how to perform a principled
empirical evaluation. An example of an empirical evaluation was present in a previous version of the paper,
but it was cut for space. It is clear that, for the type of evaluation we advocate to be better understood and
adopted, more description and guidance is needed. We will include a detailed example in any revised version.

**Interventional vs Structural measures on the correct structure (R2)**  Reviewer 2 is correct that, for a correctly
learned structure, as long as the parameterization is correct, interventional and structural measures should
agree. Interventional measures are valuable when the learned structure is only approximately correct, and
empirical evidence indicates that this is almost always the case. We will add more discussion of this to any
revised version to make this clearer. Some related experiments were moved for space to the Supplemental
Material, showing how structural Hamming distance and structural intervention distance penalize over- and
under-specification differently than total variation distance.

**Untested influences (R2)**  The critique of untested influences refers to potential "unknown unknowns" in the data-
generating system. In many real-world systems, even with strong domain knowledge, there often exist factors
that are outside the researcher's knowledge. This is generally not possible in a synthetic system. While latent
variables can be added, they are still defined and created by the researcher, limiting the realism of the data. We
will clarify this in any revised version.

**Number of samples in synthetic data experiments (R2)**  The number of samples was chosen to match the number of
samples available from the software system. This is 2599 for networking, 473 for jdk, and 5000 for postgres.
We acknowledge there may be issues with low sample sizes and will discuss this in any revised version.

**Structural measures implicity assume a DAG (R2)**  Our intent with this statement was that some structural measures
can only be used by algorithms that produce a DAG as output. There are many ways this assumption could
be violated. Acyclicity is one such violation, which could hinder the use of certain structural measures in
evaluation of an algorithm that outputs a cyclic graph. Another possible violation is an algorithm that does not
output a graphical model, such as a recent work on learning probabilistic programs. We will clarify this intent
in any future version.

**Total variation distance calculation (R2)**  In Figure 2, TVD is calculated as the distance between the learned distribu-
tion and the true distribution, just as SHD is calculated as the distance between the learned structure and the
true structure. Here, the true distribution is known because the data is synthetically generated, although such
true distributions can also be obtained from interventional experiments as they are elsewhere in the paper.

**Domain-specific simulations (R3)**  Domain-specific simulation systems are a very useful approach to consider, and
we will add a discussion of them in any future version of the paper. We already mention one such system (the
DREAM in-silico challenges), but we will add a far more general discussion of this approach. A sufficiently
sophisticated simulation falls on a spectrum between purely synthetic and purely empirical data, and such
simulations are valuable because they are often highly complex, are created by someone other than the
researcher, and are created for a purpose other than evaluation.

**Capabilities and limitations of current interventional datasets (R3)**  We agree that a more detailed discussion of
the currently available interventional datasets would be highly beneficial. This sort of discussion was omitted
for space but will be included in at least the Supplemental Material of any revised version.

[Meta-Review · NeurIPS 2019]

The reviewers agreed that this paper addresses an important notion that should be disseminated widely in the ML community working on causal learning. While some reviewers were concerned that sample size issues may lie at the root of some of the findings of the paper, most found that the papers' contribution is more foundational: is asks what types of questions and metrics should even be used when evaluating causal inference methods. Beyond the wide survey of existing practice, the proposal for interventional measures and the novel type of benchmark dataset proposed would be interesting and useful to the community.